# How Chaotic Is Genome Chaos?

**DOI:** 10.3390/cancers13061358

**Published:** 2021-03-17

**Authors:** James A. Shapiro

**Affiliations:** Department of Biochemistry and Molecular Biology, University of Chicago, Chicago, IL 60637, USA; jsha@uchicago.edu

**Keywords:** DNA break repair, alternative end-joining (alt-EJ), chromothripsis, chromoplexy, chromoanasynthesis, retrotransposition, target-primed reverse transcription (TPRT), immunoglobulin VDJ joining, class switch recombination (CSR), human papillomavirus (HPV)

## Abstract

**Simple Summary:**

Cancer genomes can undergo major restructurings involving many chromosomal locations at key stages in tumor development. This restructuring process has been designated “genome chaos” by some authors. In order to examine how chaotic cancer genome restructuring may be, the cell and molecular processes for DNA restructuring are reviewed. Examination of the action of these processes in various cancers reveals a degree of specificity that indicates genome restructuring may be sufficiently reproducible to enable possible therapies that interrupt tumor progression to more lethal forms.

**Abstract:**

Cancer genomes evolve in a punctuated manner during tumor evolution. Abrupt genome restructuring at key steps in this evolution has been called “genome chaos.” To answer whether widespread genome change is truly chaotic, this review (i) summarizes the limited number of cell and molecular systems that execute genome restructuring, (ii) describes the characteristic signatures of DNA changes that result from activity of those systems, and (iii) examines two cases where genome restructuring is determined to a significant degree by cell type or viral infection. The conclusion is that many restructured cancer genomes display sufficiently unchaotic signatures to identify the cellular systems responsible for major oncogenic transitions, thereby identifying possible targets for therapies to inhibit tumor progression to greater aggressiveness.

## 1. Introduction

There is growing recognition that cancer progression is fundamentally an evolutionary process [1,2,3]. Like cellular and organismal evolution, cancer evolution occurs largely in a punctuated, “macroevolutionary” manner, with major genome changes frequently occurring at key transition stages (e.g., initiation, tumor organoid formation, transition from benign to malignant, metastasis, acquiring resistance to anti-tumor therapies) [1,4], 

Our 21st Century understanding of cell and organism evolution is based in part on well-documented examples of ecologically triggered action by a wide variety of infective and intracellular “natural genetic engineering” (NGE) systems, such as many classes of mobile DNA elements and viruses that integrate into and modify host cell genomes [5,6,7]. Such genome change operators allow the organisms/cells to survive ecological stress and reconstruct their genomes to adapt to novel circumstances. In many cases, we can trace the action of specific classes of NGE agents in major evolutionary innovations, such as viviparous reproduction in mammals [8]. Can we find evidence for the operation of such NGE systems in cancers and recognize patterns that may lead to possible therapeutic interventions?

Heng and colleagues have characterized the rapid genome restructuring episodes leading to tumor cells with novel adaptive capabilities [9] as “Genome Chaos” [1,10]. Although intended to signify a process of genome reorganization under crisis with a high degree of heterogeneity, the word “chaos” suggests unpredictable and disorganized rapid genome modification at moments when cancer cells experience extreme stress. Cancer genomics is in the early stages of examining how stress-driven restructuring occurs. It seems logical, therefore, to pose the question that is the title of this article: how chaotic is “Genome Chaos”? Do cancer genomes change in a random and stochastic manner, or is there evidence for more well-characterized and predictable genome restructuring processes at work in these complex episodes? 

As a microbial geneticist relatively new to cancer genomics, it seems to me that “Genome Chaos” is probably less chaotic than its name implies. Just as organismal evolution utilizes a broad but not unlimited repertoire of natural genetic engineering capabilities, it may well prove to be the same for cancer evolution. Organisms have continually needed to evolve reliable genome repair and change processes to be able to meet the demands of changing conditions, and the same applies to tumors. There are at least three non-random aspects of oncogenesis I have encountered in my non-specialist review of tumor genomics that I believe point towards highly diverse but still organized, not chaotic, restructuring processes in cancer genome evolution: A limited number of distinct, well-defined and highly evolved genome repair and restructuring processes that generate characteristic and recurring molecular signatures in many cancers.A recurring spectrum of recognizable major genome restructuring signatures which vary from cancer to cancer [11].The importance that cell type or virus infection history plays in stimulating characteristic types of genome change in specific cancers by regulating the genome restructuring processes that operate in those tumor cells.

Let us look at these three points in more detail. It is fortunate that a large number of papers relevant to this topic have recently emerged from the Pan-Cancer Analysis of Whole Genomes, comprising detailed sequence analysis of over 2600 whole cancer. genomes [12,13,14]. The results in those and earlier papers provide specific examples of how a limited number of highly evolved genome repair and restructuring systems can quickly produce the enormous variety of large-scale genome changes observed in both cancer and organismal evolution. As one author puts it, cancer is “evolution within a lifetime” [15].

## 2. Published Results on Genome Structural Changes in Cancer

### 2.1. A Limited Number of Distinct Highly Evolved Genome Repair and Restructuring Processes Operate in Many Cancers

Human cells possess a reasonably well-defined set of replication, repair and mobile DNA systems that carry out the great majority of complex genome restructuring events found in cancer. 

(a)Three distinct DNA repair networks capable of joining together and rearranging broken chromosomes are found among eukaryotes from *Saccharomyces* to plants and animals [16,17,18]. These networks carry out homology-dependent repair (homologous recombination = HR and single-strand annealing = SSA), non-homologous end-joining (NHEJ), and an alternative end-joining (alt-EJ) process involving DNA replication that has received various designations in the literature over the years, including “microhomology-mediated break-induced replication” (MMBIR) and “Theta-mediated end-joining” (TMEJ) [19,20,21,22,23,24].HR generally leads to error-free recombinational break repair templated on the undamaged homologous chromosome or sister chromatid, except that HR and SSA can lead to deletions and translocations when sequence homologies are repeated at different chromosomal locations, producing “non-allelic homologous recombination” (NAHR) [18,25].NHEJ processes and joins broken chromosome ends with limited changes at the two breakpoints [26,27]. Thus, for breaks on a single chromosome, NHEJ tends to introduce only localized sequence variation, but when multiple chromosomes are broken, NHEJ can form translocations and other rearrangements without copy number increases.Alt-EJ requires the activity of the multi-functional DNA polymerase Theta (Pol θ), involves DNA synthesis with microhomology-mediated template switching as well as untemplated synthesis, and can introduce localized copy number variation (CNV) plus complex intra- and inter-chromosomal rearrangements into the repair junctions [23,24,28,29].(b)In addition to these three basic cellular DS break repair systems, human cells contain two additional DNA joining systems:In the human genome, the only normally active mobile DNA elements are the “long interspersed nucleotide elements” (*LINE1*s), which encode the reverse transcriptase and endonuclease activities needed for “target-primed reverse transcription” (TPRT) and integration at new genomic locations [30,31]. *LINE1*-encoded activities also function to mobilize “short interspersed nucleotide elements,” (SINEs), which do not have protein-coding capacity. In the human genome, the major SINE element is *Alu* (>1,500,000 copies), but another significant hominid-specific somatically active SINE group is the family of composite elements that go by the designation *SVA* (*SINE*/variable number of tandem repeats/*Alu*) (>2762 copies) [32]. *LINE1*-mediated TPRT involves two DNA-joining events: (i) endonuclease cleavage and polymerization from the exposed 3’-OH group at the target site to initiate cDNA synthesis (the “target priming” step) and (ii) ligation of the 3’-OH at the end of the cDNA strand to a 5’-PO_4_ group exposed in a target DNA DS break to terminate reverse transcription [33]. These two steps are followed by synthesis of the missing complementary strand to complete insertion of retrotransposed DNA between the initiation and termination sites. When these events occur at closely spaced endonuclease cleavage sites on target DNA, the result is an insertion flanked by a short 5 bp “target site duplication” (TSD). However, the two ligations may also occur at more distant sites on the target DNA or the 3’ cDNA insertion in an endonuclease-independent manner at the end of a pre-existing DNA break site [34]. When TPRT utilizes such exceptional distant target ligation sites to initiate and terminate reverse transcription, chromosomal structural variations can occur, in particular deletions and translocations, inversions, duplications and chromosome fusions [34,35,36]. The vast majority of these structural changes carry a signature of L1 TPRT in the form of retrotransposed polynucleotides bridging the rearrangement breakpoints.Adaptive immune system DNA changes. Restricted to lymphatic tissues and their cancers, where B cell lymphoma accounts for 95% of all lymphomas [37], these dedicated natural genetic engineering activities have evolved to generate antibody diversity by VDJ joining, antibody affinity maturation by somatic hypermutation (SH), and targeting of high affinity antibodies to different tissues by heavy chain class switch recombination (CSR) [38,39,40,41,42,43,44]. In VDJ joining to construct the variable region exons encoding the amino-terminal portion of antibody light and heavy chains are assembled by controlled chromosome breakage and joining reactions in primary B cells. The RAG1 plus RAG2 proteins making the breaks (with characteristic hairpin ends) evolved from a DNA transposase [41,42]. They cleave variable region-coding chromosomal DNA in a precise order at specific “recombination signal sequences” (RSSs) on the boundaries of V, D, and J coding cassettes so that the broken ends can be joined to form VJ light chain exons and VDJ heavy chain exons. Later in B cell development, antigen-activated B cells undergo SH in the germinal center by targeted action of “activation-induced cytosine deaminase” (AID) at the variable region exons for both heavy and light chains [45,46,47,48]. Subsequent selection of cells producing more tightly binding antibodies results in “affinity maturation” of the antibody response. Still later in B cell development, mostly outside the germinal center [49], an AID-dependent process of cytokine-targeted chromosome breakage occurs at an Sµ “switch region” upstream of the heavy chain germline Cµ region exon and also at one of the switch regions upstream of the Cγ, Cα, and Cε exons. After NHEJ or alt-EJ ligation of the two S region breaks, the “class-switch recombination” (CSR) is complete, and the new C region of the high-affinity antibody’s heavy chains targets it to an appropriate tissue in the body [50]. (c)In addition to the distinct biochemical complexes that carry out widespread DNA restructuring in cancer evolution, at least two higher-level cell biological routines that have been identified as regular programmed responses to genome damage and stress. Mitotic errors lead to micronucleus formation around lagging chromosomes or their breakage products [51,52]. Micronuclear inclusion in a daughter cell from such an incomplete mitotic division leads to chromosome fragmentation and MMBIR/alt-EJ replication in G2/M phases of the following cell cycle, with unequal distribution of the resulting partially amplified fragments to the granddaughter cells [51,52,53]. Reconstruction of a heritable linear or circular chromosome from alt-EJ joining of these fragments produces major scrambling of segments from the micronuclear chromosome with multiple copy number variations (CNVs) [54]. Cited by Heng and colleagues as a source of genome chaos [10], micronuclei are found in plants as well as animals and are associated with the DNA damage response and chromothripsis, indicating a deep evolutionary history for micronucleus formation and chromosome fragmentation among eukaryotes [55,56].Cell fusions [57], physical injury and wound-healing, or reproductive stress can all lead to the formation of non-mitosing, endoreplicating giant polyploid cells [58,59,60]. Polyploids have an old evolutionary history [61], are found in all major eukaryotic groups (plants, animals and protists), are important in plant and animal organogenesis [62] and constitute a general stress response in plants [63,64]. However, polyploidy also leads to major genomic instability [65]. The polyploid response connection to cancer has been known for over a century, at least since the pioneering cytologist Theodor Boveri linked it to human skin malignancies localized at scars and burn sites in 1914 [66]. In cancer, there are “polyploid giant cancer cells” (PGCCs), which form in response to stresses such as radiation [67], senescence [68], telomere crisis [69,70] and anticancer chemotherapy [71]. After formation and a lengthy period with DNA replication but no cell division, PGCCs ultimately undergo multipolar amitotic divisions to generate cells with smaller, dramatically restructured genomes. Some of the PGCC progeny cells have pseudo-diploid genomes and proliferate. Among these are new cancer cells with chromosome reorganizations and major steps in cancer progression, including acquisition of stem cell-like properties and immortality [72,73,74,75,76].

### 2.2. A Recurring Spectrum of Recognizable Major Genome Restructuring Signatures Which Vary from Cancer to Cancer

One of the key indicators that genome reorganization in cancer progression is not a chaotic process is that specific genome structural variation (SV) “signatures” are found in individual cancers, and these characteristic signatures are sufficiently distinct to rule out a fully chaotic or stochastic process [11,77]. We understand the mechanisms and cell biology events behind a few of these restructuring signatures.

#### 2.2.1. Chromothripsis

Chromothripsis (meaning “chromosome shattering”) involves multiple breakage-rejoining and copy-number variations (CNVs) on a single chromosome, chromosome region, or small set of chromosomes [78,79,80]. Experimental evidence links the process of chromosome shattering to mitotic errors resulting in formation of “lagging” chromosomes at mitosis, where they are encapsulated in a micronucleus envelope [52,54,81]. A micronucleus chromosome is distributed to one daughter cell at the completion of mitosis and undergoes fragmentation into actively replicating segments during the mitotic phase of the subsequent cell division cycle [20,54,82]. Later these unevenly amplified fragments can be found assembled into a scrambled chromothripsis product. The sequences in a chromothripsis structure include tandem short template jumps (TSTs) characteristic of alt-EJ repair processes [54]. As these results would lead us to expect, chromothripsis occurs independently of NHEJ [83] The visible result of chromothripsis is a set of many tightly nested intrachromosomal rearrangements on a particular chromosome (Figure S2 of [78]). Chromothripsis is also associated with mitotic errors due to “hyperploidy” or telomere damage in laboratory cell cultures and in primary medulloblastoma cancer genomes in vivo [84].

#### 2.2.2. Chromoplexy

Chromoplexy (meaning “chromosome weaving”) is found “in nearly 63% of all prostate cancer samples and 30% of all bladder cancer samples” [85]. In comparison to chromothripsis, chromoplexy involves a completely different but equally non-random rearrangement signature [86,87,88]. In chromoplexy, a small subset of chromosomes from the full karyotype undergo multiple connected or “chained” exchanges to produce translocations and other nested rearrangements without copy number variation [20]. Besides multiple exchanges being limited to a small subset of the genome (e.g., 3–5/23 chromosomes in Figure 4 of [78]), 50% of all breakpoints in a chain are closer to another breakpoint on the same chromosome than would occur by chance [86]. This spatial clustering suggests that chromoplexy involves proximity of chromosome domains. These domains appear to be determined by patterns of genome expression because there is a correlation between breakpoint sites on different chromosomes and regions known to be spatially linked in transcription factory “topologically associated domains” (TADs) [89,90]. In other words, the 3D organization of the genome in TADs for expression during interphase appears to play a role in setting up different chromosomes for rearrangement chains in chromoplexy events. From these parameters, we can deduce that chromoplexy occurs during interphase, far earlier in the cell cycle than chromothripsis, and involves non-replicative end-joining, probably NHEJ [20]. Again, the limited number of chromosomes involved and the restricted timing during the cell cycle in chromoplexy argue against a chaotic genome restructuring process. 

#### 2.2.3. Chromoanasynthesis

Chromoanasynthesis (meaning “chromosome restitution” or “chromosome building up”) is a process that produces a series of clustered, amplified and rearranged chromosome segments inserted into one or a few discrete loci on a single chromosome. These “focal CNVs” arise from broken replication forks by MMBIR/alt-EJ repair or a related process at gapped replication sites known as “fork stalling and template switching” (FoSTeS): “When the replication fork stalls or pauses in stressed cells, the lagging strand can serially disengage and switch to another nearby active replication forks, leading to the template driven joining of several sequences from different genomic regions, before the resumption of the replication on the original template” [91]. In both cases, the same regions can undergo replication several times and form palindromic repeats due to foldback templating. Chromoanasynthesis has been found to account for CNV of oncogenic loci in breast cancer [92], B-cell lymphoma [93], mouse brain tumors [94], and on three separate chromosomes, 6, 7 and 12 in a renal leiomyosarcoma [95]. There are also examples of congenital germline chromoanasynthesis on human chromosomes 1 [96] and 21 [97] without obvious pathology but deleterious effects on chromosome 9 [98], 11 [99], 13 [100], 14 [101], and 18 [102].

A special form of chromoanasynthesis occurs in triple-negative breast cancer and in ovarian, endometrial, and liver cancers that originate from individuals with a genetic defect affecting the homologous recombination protein BRCA1, or in clones that have lost BRCA1 and TP53 by mutation. In these TP53-BRCA1-deficient cells, sequencing has identified the tandem duplicator phenotype (TDP) characterized by an enrichment of head-to-tail segmental tandem duplications (TDs) distributed across the genome [103,104,105]. Most of the tandem duplications fall into two normally distributed size classes, 1.6–51 kb (mean = 11 kb) and 51–622 kb (mean = 221 kb), with opposite effects on expression of the duplicated segments: the shorter TDs disrupt and inactivate expression of tumor suppressor loci, while the longer TDs duplicate oncogene loci or their transcriptional regulatory sequences to enhance expression [104,106]. The difference between the two size classes is activation of the CCNE1/cyclin E1 pathway in tumors with the longer TDs. [106]

In cells lacking BRCA1 and TP53, broken chromosome ends at stalled replication forks appear to be replicated and joined by an alt-EJ process that involves DNA polymerase θ rather than by canonical NHEJ [29,106,107,108]. Pol θ has helicase activity as well as microhomology-directed polymerase template switching and terminal transferase activities that insert extra nucleotides, sometimes untemplated, to leave characteristic “scars” between the tandemly joined sequences [23,109,110]. The role of Pol θ in forming TDs by error-prone alt-EJ has been confirmed by knockout studies in cancer tissue cultures [111] and supported by congruent results in the genetically tractable nematode worm, *C. elegans* [112,113].

#### 2.2.4. Double Minute Chromosomes/Extrachromosomal Circular ecDNAs

Sometimes chromothriptic fragments reassemble into extrachromosomal circular “ecDNAs” or “Double minute” (DM) chromosomes, generally lacking centromeres [114,115]. Recently “signatures of chromothripsis in 36% of circular amplicons…and half of circular amplicon cases” across multiple cancers were reported, confirming “recent observations that chromothripsis can result in ecDNA formation” [52,54,116,117]. 

Lacking centromeres, ecDNA/DMs are subject to unequal distribution at cell division. Consequently, ecDNAs can amplify tumor-promoting “oncogenes” in cancer cells for at least three reasons: (i) MMBIR/alt-EJ produced extra copies during the original chromothripsis event, (ii) they carry sequences permitting episomal replication, and/or (iii) they accumulate in particular lineages of progeny cells as a result of biased segregation. Copy-number increase by biased distribution at cell division also leads to tumor heterogeneity, as frequently observed in aggressive late-stage tumors. 

If more than one chromosome undergoes chromothripsis, ecDNAs can form containing segments from two or more chromosomes with junctional microhomologies indicating assembly by alt-EJ mechanisms [116,117,118]. ecDNAs can also recombine to produce tandem intramolecular amplifications and reintegrate into multiple chromosomal locations to form what are called “homogeneous staining regions” when visualized with oncogene in situ hybridization probes [114,118,119]. An alternative origin for ecDNAs/DMs containing sequences from more than one chromosome involves recombination, fusion and rearrangements between different ecDNAs, each arising from a distinct chromosome [120].

Almost half of late stage tumors contain amplified oncogenes on ecDNA, which contain the most common focal amplifications in cancer [121]. The presence of ecDNA in tumors is often an indicator of an advanced stage in cancer evolution and a poor clinical prognosis [116]. Oncogenes on ecDNA are more actively transcribed than they are from chromosomal amplifications in part because ecDNA chromatin is more open than normal chromatin and “ecDNA is shown to have a significantly greater number of ultra-long-range interactions with active chromatin” [115,122].

#### 2.2.5. LINE1-Mediated Retrotransductions and Large-Scale Genome Rearrangements

*LINE1* retrotransposition occurs regularly in human pluripotent stem cells and nervous system development [123,124]. There are >850,000 copies of *LINE1* elements in the human genome (~17% of total DNA), but only ~100–150 are actively mobile, and about 20 “hot L1s” carry out the vast majority of retrotransposition events [125,126,127]. Recurrent *LINE1* insertions into the *APC* locus are thought to initiate colorectal tumor development [128]. 

In addition to LINE and SINE mobilization, *LINE1*-encoded TPRT requires only a 3’ poly-A_n_ tail on the template RNA [129]. That almost complete lack of sequence specificity allows TPRT to mediate genome insertion of cDNAs from several classes of RNA:TPRT mobilizes read-through 3’ transcripts downstream from the original *LINE1* element (“L1 transduction”) and provides a molecular mechanism for exon shuffling in the human genome [130]. “The 3’ transductions disseminated genes, exons, and regulatory elements to new locations, most often to heterochromatic regions of the genome” [131]. A recent report states that 24% of all retrotransposition events in cancer contain a 3’ transduced segment [128].A parallel process mobilizes 5’ read-through transcripts upstream of SVA SINE elements [132,133].*LINE1* TPRT can insert cDNAs from spliced mRNAs and some stable RNAs, such as U6snRNA, into the genome and generate “processed pseudogenes” or “retrogenes” [134,135,136]. These processed pseudogenes are often expressed and can play a role in oncogene amplification during cancer [137].There are reports of 20 deletions containing insertions of *Alu* and *SVA* SINE elements in human germline, somatic and cancer breakpoint junctions [138,139,140], Further, the Pan-Cancer Genome Study has recently published a detailed analysis of exactly such *LINE1*-mediated genome restructuring in “2954 cancer genomes from 38 histological cancer subtypes“ [141]. Intriguingly, the distribution of *LINE1*-mediated TPRT events is far from randomly distributed among different cancers. While ~50% of tumors show some evidence of LINE1 retrotransposition, only four cancer types predominately displayed >10 events per genome: esophageal adenocarcinoma (~75% of samples), head squamous cell carcinoma (~35%), lung squamous cell carcinoma (~55%) and colorectal adenocarcinoma (~40%). Altogether these four tumor types contained 70% (13,373/19,166) of all TPRT events among the entire PCAWG dataset but constituted just 9% (266/2954) of the samples. In esophageal adenocarcinoma, 27% of the samples contain more than 100 separate TPRT events, which are the most frequent type of structural variation in this cancer. TPRT generates the second-most frequent type of structural variants in head-and-neck squamous cell and colorectal adenocarcinomas. Of the remaining cancer types showing >10 TPRT events in some fraction of tumor genomes, 9 were classified as adenocarcinomas and 1 each as squamous cell carcinoma, chronic lymphocytic leukemia, pancreatic endocrine cancer and hepatocellular carcinoma. The PCAWG dataset includes the following signature features and rearrangements attributable to *LINE1* TPRT-mediated SVs:A 244-fold enrichment of target site selection for those containing close homology to the preferred consensus sequence (5′-TTTT/A-3′) for L1 endonuclease cleavage and polyA tail hybridization [141].An 8.9-fold preference for insertion into late-replicating regions of the genome near the end of S phase in the cell cycle, in agreement with earlier work [142].Five hot LINE1 elements account for 50% of all transductions. LINE1-mediated chromosome deletions account for −1 CNVs observed adjacent to 5′ TPRT insertion sites. Evidence for the TPRT mechanism includes the presence of retrotransposed sequences bridging deletion breakpoints and homology to the L1 endonuclease target preference at the 3’ end of the insertion/deletion event in the cancer cells. Similar insertion/deletions occur in healthy human brains [143]. Intrachromosomal deletions range in size from 0.1 kb up to ~80 Mb. Notable examples include an esophageal tumor with a 45.5-Mb interstitial deletion of chromosome 1 flanked by TPRT signatures and a lung tumor with an L1-mediated deletion of 51.1 Mb from chromosome X, including the centromere [141].Inter-chromosomal translocations connected by TPRT-generated bridging sequences [141].Terminal deletions and isochromosome fusions leading to breakage-fusion-bridge (BFB) cycles, as first described by McClintock, leading to multiple CNVs [144]. The terminal fusions would occur by a TPRT bridge initiating at a sub-telomeric site on one sister chromatid and terminating at a break site on the same arm of the other chromatid. This sequence of post-replication events on Chromosome 11 led to loss of 50 Mb of the telomere end of 11q in a lung cancer and loss of 53 Mb of the same chromosome arm in an esophageal adenocarcinoma, combined in both cases with amplification of the *CCND1* oncogene encoding cyclin D [141].A tandem 22.6 Mb duplication on the long arm of Chromosome 6 in the same esophageal adenocarcinoma increased the copy number of the *CCNC* locus encoding cyclin C, also a putative oncogenic protein [141]. A fold-back inverted 79.6 Mb duplication formed by end-to-end joining of sister chromatids on Chromosome 14 occurred in a lung tumor [141].

## 3. The Importance of Cell Type or Virus Infection History in Characteristic Types of Genome Change in Specific Cancers

### 3.1. B Cell Lymphomas and Leukemias

One of the most striking pieces of evidence for questioning the chaotic nature of genome restructuring in cancer are the observations that cancers derived from B lymphocytes frequently contain rearrangements involving proteins and DNA substrates used for immunoglobulin synthesis and maturation. The involvement of the immunoglobulin loci *IGH* on chromosome 14, *IGK* on chromosome 2, and *IGL* on chromosome 22 as hotspots for recurring translocations activating different oncogenes has been evident for many years [145,146] Genetic experiments in mice provided evidence that the RAG2 subunit of the VDJ recombinase is required for oncogenic translocations [147]. A recent paper characterized the frequency and locations of inter-chromosomal oncogenic rearrangements involving *IGH*, *IgK* and *IgL* in different B cell-derived tumors: 6.5% for chronic lymphocytic leukemia (CLL), 98% for mantle cell lymphoma (MCL), 50% for multiple myeloma (MM), and 47% for diffuse large B cell lymphoma (DLBCL) [44]. Many of the following rearrangements were recurring both at the genetic locus target level and across tumor types:7 examples of *IGH*-*BCL2* in CLL;62 examples of *CCND1*-*IGH* in MCL;7 examples of *CCND1*-*IGH*, 4 of *MYC*-*IG*), and 2 of *NSD2*- *IGH* in MM;22 *IGH*-*BCL2* and 5 *BCL6*-*IGH* in DLBCL.

The recurrences reflect both the repeated action of the immune recombination system as well as the positively selective oncogenic effects of the resulting genetic fusions [148]. In particular, the *IGH* locus has several strong 3’ enhancer sites that would elevate expression of fusion oncogenes [149].

Sequence analysis has also detected deletions outside the *IG* loci that involved breakpoint sequences with significant homology to the RSS signals for VDJ joining [150,151]. About 25% of acute lymphocytic leukemias (ALLs) carry a particular genetic fusion (*ETV6*-*RUNX1*) acquired in utero, and they display ongoing and recurring deletions at multiple genomic locations that join RSS homologues throughout the course of the disease [39]. These deletions remove functions for normal B-lymphocyte development and differentiation to exacerbate the leukemia. As the authors of this ALL study wrote, “The RAG-mediated signature is unparalleled among cancer-associated mutational processes for its specificity in inactivating the very genes that would usually promote normal cellular differentiation.”

There are also cases where the AID-dependent CSR joining system played a role in mediating translocations [152,153]. The genetic hallmark of Burkitt’s lymphoma (BL) is a *MYC-IGH* fusion resulting from translocations between chromosomes 8 (*Myc*) and 14 (*IGH*) [154]. Most of these fusions occur at heavy chain class switch recombination (CSR) sites rather than VDJ joining positions. In one study, 77% of the recurring translocations on chromosome 8 occurred at two Myc clusters 560 bp and 779 bp in length within a 4555 bp breakpoint region. Intriguingly, where the Myc breakpoint occurred correlated significantly with which of the eight CSR “switch region” sites was involved: translocations at one of four *IGH* S*γ* regions occurred largely in the 5’ 560 bp cluster, the two *IGH* S*α* regions connected largely to the 3’ 779 bp cluster, while the Sµ germline region joined with a modest 5’ bias across the entire 4.5 kb *Myc* breakpoint region. Transcription across the eight different *IGH* switch regions is regulated to incorporate signals from the infection response in determining which class of antibody to produce. Therefore, it is likely that some feature of the regulatory status of each lymphoma’s B cell progenitor influenced the accessibility of the two *Myc* clusters to CSR beakage and joining. The same *Myc* breakpoint region on chromosome 8 is also a recurring hotspot of *MYC-IGH* fusions in DLBCL lymphomas [155].

Many B cell tumors also showed evidence of ectopic AID-induced somatic hypermutation [156,157,158,159]. Many of the hypermutated regions do not encode proteins but include altered regulatory sites that stimulate expression of putative oncogenic loci in DLBCL tumors [160,161]. It was known from work on immunoglobulin SHM that AID activity was targeted specifically to V region exons in the producing cells by special “diversity activation” (DIVAC) enhancer elements at each *IG* locus [162]. It has recently been discovered that these DIVAC enhancers target AID activity to execute SHM at non- *IG* sites in specific inter-chromosomal TADs in the activated B cell nucleus [163]. Moving a DIVAC element to a TAD where SHM does not normally occur activates AID in that TAD.

### 3.2. Virus-Mediated Oncogenesis and Genome Restructuring

Many different kinds of viruses have oncogenic potential, as first discovered in poultry by Peyton Rous in 1910 [164]. The Rous Sarcoma Virus (RSV) turned out to be a retrovirus capable of inserting into the chicken genome and expressing the *Src* oncogene [165,166]. Many other “tumor viruses” have since been discovered, and some of them have been shown to integrate into the human genome and directly participate in oncogenic structural variations. By so doing, they introduce genome elements not present in uninfected cells, which can be foci of genome restructuring leading to cancer development. The best-documented of these genome-modifying tumor viruses are the human papillomaviruses (HPVs) present in over 99% of cervical cancers, and integrated into the chromosomes of at least 82% of cervical tumors and 70% of oropharyngeal squamous cell carcinomas [167,168]. The integration events occur throughout both the human and HPV genomes guided by microhomologies [169]. Many of these cancers also carry non-integrated circular “episomal” copies of replication-competent HPV genomes as well as replicating hybrid episomes containing both host and viral sequences [170].

Integrated and episomal HPVs influence tumor development and genome restructuring in at least four different ways:HPVs encode two regulatory factors, E6 and E7, which act as “oncoproteins” to stimulate cancer cell growth: “The HPV-infected cells maintain their proliferative potential and remain uncoupled from differentiation through the inactivation of key cell cycle regulators, including members of the pRb family of proteins, pRb, p107 and p130. The HPV E7 proteins interact with these essential cell cycle regulators, leading to the release and activation of E2F transcription factors that regulate S phase genes… E6 proteins have evolved to target p53 for proteasomal-mediated degradation…This cooperative action of the high-risk E6 and E7 oncoproteins abrogates multiple cell cycle checkpoints, thereby allowing genome amplification whilst ensuring the continued survival of the infected cell… E6 induces the hTERT promoter via interactions with c-Myc and NFX-1 proteins and contributes to cellular immortalization…” [171].The disruptions of cell cycle control by E6 and E7 proteins lead to mitotic errors and activate cellular DNA repair functions, whose expression is necessary to enable viral replication [171,172,173,174,175,176]. HPV replication specifically requires the homologous recombination functions RAD51 and BRCA1 [177,178], even though it has been reported that E6 and E7 “impair the homologous recombination pathway” [179]. E7 oncoprotein binds the host Rb protein to suppress canonical nonhomologous end-joining (NHEJ) and promote error-prone alt-EJ [169,180]. These changes modify the pattern of genome structural variations and facilitate viral insertion into the host cell genome.HPV genomic and sub-genomic sequences participate directly in generating “focal” structural variations of human chromosomes clustered near the original insertion site. Analysis of individual clones from a particular HPV-positive cervical or head-and-neck tumor indicated that viral CNVs occurred following an initial insertion event [181]. Sequence analysis of regions up to 3 MB long displaying host genome CNVs found that the additional copies of human DNA were invariably flanked by HPV sequences, whether they were deletions or amplifications. Similarly, chromosome rearrangements contained amplified HPV sequences at the breakpoints: “Statistical analysis confirmed a strong association between HPV insertional breakpoints and genomic structural variation in eight of nine cases, including chromosomal translocations, deletions, inversions, and/or intrachromosomal rearrangements (Bonferroni adjusted binomial test, all *p* < 10^−9^) … Moreover, such enrichment occurred at focal hyperamplification sites in six of nine samples (local ploidy >8 N, Bonferroni adjusted binomial test, all *p* < 10^−10^)” [181]. Apparently, the HPV sequences provide a first site for chromosome breakage initiating excision and replication of hybrid host-viral segments that can then be reintegrated adjacent to the original insertion (in the case of local CNVs) or at a new genomic site (in the case of larger scale SVs). Many chromosome regions with local CNVs contain several interspersed HPV-host segments, indicating the clustered occurrence of multiple excision-reinsertion events.Using HPV replication proteins E1 and E2, integrated HPV fragments act as initiation sites for focal bidirectional “onion skin” amplifications and CNVs of surrounding chromosomal DNA [182]. The products of this in situ replication may be joined together to create local tandem repeats, but they can also be joined with fragments from other chromosomes to form hybrid viral-multi-chromosomal fusions, frequently found as episomal ecDNA replicating in cervical cancer samples [170]. Both tandem repeat and episomal fusion structures have been found to serve as the basis for oncogene amplification. In addition, integrated HPV fragments can multimerize enhancer elements to produce super-enhancers that drive high levels of viral oncogene transcription [183].

## 4. Discussion

Rapid multi-site genome restructuring marks many stages in cancer development [1,78,184]. These episodes of abrupt genomic transformations have been dubbed “genome chaos” to denote the breakdown of normal controls on genome stability and the complexity of many SVs in the new genomic configuration [1,10]. The term suggests a random process of change. However, a closer look at the cellular processes that lead to cancer cell genome restructuring reveals a number of strikingly non-random unchaotic properties that indicate the operation of defined DNA rearrangement functionalities, many with long evolutionary histories in eukaryotic biology [68,185]. 

First among the non-random features of episodic cancer genome transformations are the signature structural differences discussed in the previous section. These signatures result from the operation of specific DNA restructuring systems, some of which alter chromosomes at quite different times during the cell cycle: chromoplexy in interphase [20], *LINE1* TPRT in late S phase [142], and chromothripsis in late mitosis [54]. 

Next is the long-recognized impact of the history of viral infections on genome changes in cancer [186]. As we have seen in the best-studied case of HPV, for example, the virus provides both regulatory proteins directing DNA repair systems to mutagenic microhomology-mediated events as well as DNA substrates for episomal replication, focal rearrangements and CNVs [168,169,181]. 

Finally, there is the tumor type- and tissue-specificity of genome restructuring processes. This is most obvious in the case of lymphoid tumors derived from B cells that virtually uniquely utilize adaptive immune system DNA restructuring functions to produce oncogenic rearrangements and introduce SHM changes into particular sites in the evolving cancer genome. Although lymphomas and leukemias are unique with respect to immune system activities, all genome restructuring processes show major differences in frequencies between cancers from different tissues. The fact that only four out of 38 tumor types examined by the PCAWG Consortium contain 70% of the *LINE1* TPRT events while many other cancers contain ≤ 1 is similar to what has been found for chromothripsis and chromoplexy [14,141]. 

## 5. Conclusions

In summary, the genomic signatures in different tumors indicate that identifiable cellular and molecular processes are responsible for major hereditary variation in the course of cancer development. Delineating the unchaotic, non-random features of cancer genome restructuring is more than an academic exercise. There may well be practical therapeutic utility in identifying these functionalities for specific cancers because punctuated genome change often accompanies the most dangerous transitions of tumors towards malignancy, metastasis, and chemotherapy resistance [2,86,187]. It would therefore be beneficial to discover and apply therapies designed to inhibit such transitions. A major advantage in looking at cancer as an evolutionary process of active self-modification rather than a series of stochastic accidents could be the ability to anticipate and prevent the conversion of a tolerable disease into a lethal malignancy.

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
