# Peer review of "How Chaotic Is Genome Chaos?"

_cancers, 2021, doi:10.3390/cancers13061358_

Round 1
Reviewer 1 Report
The author has reviewed the current knowledge on abrupt genome restructuring called “genome chaos”. He summarizes the limited number of genome restructuring systems describing characteristic signatures of DNA changes resulting from the activity of these systems. He concludes that these signatures are unchaotic which may help to identify targets to inhibit or reverse tumor progression.
In summary, this comprehensive review recapitulates the current knowledge on intrinsic cellular mechanisms of genome restructuring plausibly showing that genomic changes resulting from these mechanisms are not chaotic. Some further aspects which seem to be relevant to this topic should also be mentioned by the author:
(1) All observations with respect to abrupt genome restructuring and “genome chaos” describe clonal aberrations, i.e. they are based on a minimum set of tumor cells with the same genome alteration(s), requiring that the changes in DNA / genes / chromosomes are not lethal to the tumor cell(s) and do not impede its / their cell division nor its / their expansion within the surrounding tissue. Therefore, even if the event that induced genome restructuring had occurred by chance, observations on tumor cells with clonal genome aberrations including abrupt genome restructuring cannot be “random”, they are the result of a selection process excluding all those (tumor cells with) gene / chromosome / genome aberrations which are lethal, impede cell division or tumor expansion.
(2) All tumor cells require a minimum set of genes, parts of chromosomes / of the genome relevant to protein production, cellular survival, cell division etc. working effectively. Stability / persistence of these genetic systems strongly argue against a randomness of “genome chaos”, and therapeutic approaches attacking this “Achilles’ heel” of tumor cells may also be interesting. Therefore, it would be interesting to readers of this review to get some information on the (limited) current knowledge of those parts of the genome / chromosomes / genes usually unchanged in (the great majority of) tumors even in the case of an abrupt genome restructuring (in addition to those genes responsible to genome restructuring).
Author Response
Point (1) - The MS addresses mechanisms of genome change, not the specific functions of the altered regions, except as they are related to tumor progression. I do not see a need for change in regard to the reviewer's concern for selection against deleterious mutations.
Point (2) - This recommendation is outside my area of expertise and is not relevant to the focus of the MS.
Reviewer 2 Report
In this timely piece, Professor Shapiro has asked and then addressed a very important question in current cancer evolutionary studies: How chaotic is genome chaos? Given the new observations that genome reorganization plays a key role in cancer evolution, understanding the specific mechanisms of genome reorganization is highly important, both for basic research and potential clinical implications.
In this focused review, Professor Shapiro has first highlighted the toolbox that is essential for genome repair, then compared some typical subtypes of genome re-organizations detected in cancer. Finally, he included two case studies illustrating the relationship between cell type, viral infection, and genome reorganization in cancer. Altogether, Professor Shapiro concluded that many restructured cancer genomes display sufficiently unchaotic signatures, suggesting the possibility of manipulating these cellular mechanisms for fighting cancer.
Overall, this is an excellent review discussing an important issue by illustrating different mechanisms and signatures of cancer genome reorganization. It will be welcomed by the research community.
There are two minor suggestions to further improve its presentation. First, most current sequencing data for cancer genomes are obtained from the end products of cancer evolution. It is likely that the degree of chaotic alterations is much higher during the initial stage of the macroevolutionary process. For example, as illustrated by Liu et al, genome chaos: survival strategy during crisis, the end products often are much less chaotic (Cell Cycle, 2014;13(4):528-37). Second, as the author clearly illustrated, many tools and different types of mechanisms can be involved in the process to create new genomes. Therefore, it would be challenging to predict which mechanism among a handful will be selected by evolution, especially when dealing with a drug treatment that can induce massive genome changes, where different cells can display different mechanisms.
Author Response
I appreciate the reviewer's positive comments. Both recommendations on the cellular processes leading to genome restructuring and their relevance to evolution will be addressed separately in a paper to be published in Progress in Biophysics and Molecular Biology, based on the Cancer and Evolution Symposium from Oct. 14-16, 2020. In that paper I discuss the work of Prof. Liu and other scientists as they followed cancer cell responses to stress. I also indicate how relevant the cancer results are for evolutionary biology because similar genome restructurings are found in human germlines and during evolution of other eukaryotes.
Reviewer 3 Report
This is an interesting review paper with potentially useful characterizations of the genomic transitions by which cancer cells escape chemotherapy.
I would take issue with some aspects of how the issues are framed, exacerbated by some unfortunate conventions describing the evolutionary process of cancers. In systems dynamics the expression “chaos” has a specific meaning. It implies linked processes that proceed deterministically but are linked in such a way that there are thresholds such that large changes in the state of the system appear apparently spontaneously. One problem with applying the term “chaos” to biological evolution in general or cancer evolution in particular is that neither is, at root, a deterministic process. Rather the underlying microscopic events in evolution are mutations, which appear most often due to a proton tunneling event during DNA replication. Proton tunneling, like other quantum events, is stochastic. Mutation frequency under defined conditions can be predicted, but when a particular mutation will occur can’t. A particular well-studied example in cancer is the escape of metastatic prostate cancer from androgen deprivation therapy. Further advance of the cancer can be prevented for an average of several years, but after some time which may be longer or shorter than average, at least one cancer cell will undergo a transformation, or combination of transformations, that confers resistance to the ADT and will multiply to become the dominant form in the population, and the cancer will advance. This appears to be a stochastic, rather than a deterministic, process, as evidenced by the variable time from individual to individual for escape to occur. So, it seems to me that a critique of “genomic chaos” is perpetuating a flawed concept, even if to oppose it.
I think the accurate description of the phenomena that the paper describes is that stages in the evolution of cancer are characterized by genomic rearrangements of significant scale and complexity. That being said, then it seems to me the right inference is that cancer evolution must be understood in the context of systems biology. Steps in the evolution of cancer do not constitute disorganization, but reorganization. Targets against the evolved cancer should be looked for in the context of the networks that have changed, and are likely to be found in the genomic elements, coding or non-coding, that are critical to those networks. This is a central theme of the paper, the author presents compelling evidence supporting it, and I agree.
The author’s assertions about similarity between organismic and cancer evolution are overstated. Whereas major chromosome reorganizations, including changes in chromosome number, are common in cancer evolution, they are relatively rare in organismic evolution. While we have 23 pairs of chromosomes, our closest relatives (the chimps and the bonobos) have just one more, 24. And they share that chromosome number with gorillas and orangutans, although their last common ancestor is even more millions of years in the past than their last common ancestor with us. Overall, evolutionary changes in chromosome numbers in mammals occur with a frequency of once every several million years. By contrast many human cancers create cells with altered chromosome count during the course of the disease, in some cases multiple times.
The more cogent point made by the author is that the genomic reorganizations emerging in cancer evolution are far from random, but rather can be classified into well-defined patterns and that characterization of those patterns should enable identification of likely drug targets.
That being said, the exploration of those patterns is far from simple. Primary mutations, changes in gene expression, and changes in phosphorylation patterns will all play a role, and each reorganization will need to be defined in all of these aspects. Further, within each pattern there will be variation, so that a significant number of cells displaying each pattern will need to be studied, and statistical methods used to distinguish between the signals associated with the patterns and the noise of irreducible biological variability between samples. In short, there are patterns, but the complexity of the patterns renders the information technology and the genomic technology associated with uncovering the most important molecular mechanisms underlying them, and therefore the identification of potential drug targets, quite daunting--not impossible, but a very serious challenge.
My suggestions:
Refine the comparison between organismal evolution and cancer evolution by pointing out that, although large scale genome reorganization does happen in organismal evolution, it is far more dominant in cancer evolution.
Refrain from using the term “chaotic” where “random” or “disorganized” would be the more accurate term.
Discuss as much as possible the actual steps that would be required to go from identification of a pattern of genome reorganization in a cancer to identification of potential drug targets. Include appropriate references to methods papers and databases.
Author Response
This review is not worthy of serious rebuttal. It reflects ignorance of genomics-based studies of evolutionary change, of cancer evolution, and mistakenly attributes use of the term chaos to me. Most particularly, the reviewer appears to be challenged in understanding the deep parallels between tumor evolution and organismal evolution. What can happen in cancer can and certainly does happen in evolution.